# GaSubtle: A New Genetic Algorithm for Generating Subtle Higher-Order Mutants

Fadi Wedyan [1,*] , Abdullah Al-Shishani [1] and Yaser Jararweh [2,*]

1   Department of Software Engineering, Faculty of Prince Al-Hussein bin Abdullah II of Information Technology, The Hashemite University, P.O. Box 330127, Zarqa 13133, Jordan; abdullah.asendarz@gmail.com
2   Department of Mathematics and Computer Science, Graduate School of Liberal Arts, McAnulty College, Duquesne University, Pittsburgh, PA 15282, USA
*   Correspondence: fadi.wedyan@hu.edu.jo (F.W.); jararwehy@duq.edu (Y.J.)

**Abstract:** Mutation testing is an effective, yet costly, testing approach, as it requires generating and running large numbers of faulty programs, called mutants. Mutation testing also suffers from a fundamental problem, which is having a large percentage of equivalent mutants. These are mutants that produce the same output as the original program, and therefore, cannot be detected. Higher-order mutation is a promising approach that can produce hard-to-detect faulty programs called subtle mutants, with a low percentage of equivalent mutants. Subtle higher-order mutants contribute a small set of the large space of mutants which grows even larger as the order of mutation becomes higher. In this paper, we developed a genetic algorithm for finding subtle higher-order mutants. The proposed approach uses a new mechanism in the crossover phase and uses five selection techniques to select mutants that go to the next generation in the genetic algorithm. We implemented a tool, called GaSubtle that automates the process of creating subtle mutants. We evaluated the proposed approach by using 10 subject programs. Our evaluation shows that the proposed crossover generates more subtle mutants than the technique used in a previous genetic algorithm with less execution time. Results vary on the selection strategies, suggesting a dependency relation with the tested code.

**Keywords:** search-based software testing; genetic algorithm; evolutionary algorithm; mutation testing; higher-order mutation; subtle mutants

## 1. Introduction

Mutation testing is a fault-based testing approach which effectively helps in producing quality test cases. The approach is systematic and highly automated. It works by creating copies of the program under test, called mutants, with seeded faults. The quality of test suites is evaluated by their ability to detect these faults, which is described in literature as killing the mutants.

Since their introduction by DeMillo et al. [1] and Hamlet [2], mutation testing has been thoroughly studied in academia. Empirical studies show that mutation testing is more effective in finding faults compared with other white-box testing approaches [3]. Andrews et al. [4] reported that faults generated with mutation operators for procedural programs are similar to real faults in evaluating test effectiveness. Therefore, mutation testing is also used for estimating the number of faults present in a given program, and comparing testing techniques for verification. Despite being originally a white-box approach, mutants can be generated from various artifacts (e.g., finite state machines, state charts, Petri net) [4].

Despite all these potentials, mutation testing has limited success in the industry. This is due to two main problems, the high cost of applying the approach and the problem of equivalent mutants. The cost of mutation is associated with the process of mutation analysis. The process requires generating and compiling a large set of mutants by performing a

simple syntactic change to the original program. The change is performed by using a transformation rule called a mutation Operator. A high number of mutants can be generated even for a simple program. The number of mutants grows with the square of program size ($N^2$ mutants for a program of N lines) [5]. These mutants are then run against the test cases. The quality of the test cases is determined by computing the mutation score, also called mutation adequacy [1].

Mutants that are not killed by any test case in the test suite are said to be alive, which suggests improving the test suite by adding more test cases to kill these mutants. However, some of the alive mutants cannot be killed by any test case because they produce the same output as the original program. Such mutants are called equivalent mutants and need to be identified and eliminated if possible. Budd and Angluin [6] proved that detecting equivalent mutants is an undecidable problem. As a result, the detection of equivalent mutants need to be performed manually. Schuler and Zeller [7] found that about 45% of the alive mutants are equivalent. Moreover, they found that manual detection of alive equivalent mutants takes about 15 min per mutation, which shows that the problem is serious and widespread.

Many approaches have been proposed to reduce the cost of mutation testing and minimize the effect of equivalent mutants. Cost-reduction techniques include the use of selective mutation [8], weak mutation [9], mutant sampling approach [10], using clustering algorithms to choose a subset of mutants [11], and strong mutation [12]. Minimizing equivalent mutants techniques include compiler optimization techniques [13], approaches using mathematical constraints to automatically detect equivalent mutants [14], using program slicing to assist in the detection of equivalent mutants [15], selective mutation [8], examining the impact of equivalent mutants on coverage [16], examining changes in coverage to distinguish equivalent mutants [7], and co-evolutionary search techniques [17].

Studies have found that the majority of mutants that contain one fault, which are called first-order mutants (FOMs), represent trivial faults that can be easily killed [5], and the majority of real life faults are complex and cannot be simulated by using FOMs [18]. Gopinath et al. [19] studied faults in 6000 programs written in various programming languages and found that real faults consists of three to four tokens that can not be simulated by using FOMs.

To simulate real complex faults, higher-order mutants (HOMs) were presented. A HOM is a mutant that has more than one injected fault. The order of mutants represent the number of injected faults. For most mutants, injecting more faults into a FOM tends to make it easier to kill. However, exceptions to this rule are very interesting [5]. A number of studies [5,20–26] show that higher-order mutation presents a solution to the problems of mutation testing as the approach can produce a small set of hard-to-kill mutants with low percentage of equivalent ones.

Some of the HOMs can be harder to kill than FOMs, because the combination of faults may present more complex new faults, as one fault may mask another. However, some are easy to kill, as the injection of multiple faults into the mutant may make it weaker. Thus, HOMs that are harder to kill should be sought; such mutants are called subtle HOMs.

The space of candidate HOMs is huge, and the percentage of easy-to-kill mutants is high; thus, the act of searching for all the subtle HOMs among the huge space of candidate HOMs is costly [23]. Therefore, a search algorithm can be used in order to find a near optimal solution.

In this work, we introduce a new genetic algorithm for generating subtle HOMs. The algorithm uses five selection strategies, and a new technique for crossover. We developed a tool to automate generating, compiling, and executing both first and higher-order mutants. The tool, which we called GaSubtle (genetic algorithm for generating subtle higher-order mutants), is easy to extend as it allows us to plug other algorithms without modifying existing code. We evaluated our algorithm by using 10 subject programs in terms of effectiveness, and cost.

Genetic algorithms are widely used for solving optimization problems. In this study, we aim at answering the following questions.

- Which of the five selection strategies is more effective in producing subtle HOMs? Effectiveness of each selection strategy is computed by using the number of subtle HOMs, the number of HOMs generated, and the distribution of the degree of the subtle HOMs.
- Which of the five selection strategies is more costly? Cost is computed by using the running time. We measured the running time of the proposed algorithm with each of the five selection strategies.
- What is the effectiveness of the proposed crossover technique, compared with single point crossover? The number of generated subtle HOMs by the single point crossover and proposed crossover (which we call GaSubtle crossover) are compared. The goal is to determine if GaSubtle crossover is more effective in finding subtle HOMs.

The rest of this paper is organized as follows. Section 2 summarizes work related to higher-order mutation testing. Section 3 describes the proposed approach for generating subtle mutants. Section 4 describes the implementation of GaSubtle tool. Section 5 describes how the empirical study was set up for evaluating the proposed approach. Section 6 presents the results of the empirical study. Section 7 describes the threats to validity. Finally, conclusions and future work are discussed in Section 8.

## 2. Related Work

HOM gained the attention of many researchers in the last decade. This is mainly due to the potential of HOM to solve the problems of mutation testing, thereby making mutation testing applicable and adoptable by the industry. Here, we briefly discuss important work on HOM testing. A recent mapping study on HOM was performed by Lima and Vergilio [27], and a systematic literature review was performed by Ghiduk et al. [28]. A systemic literature review on cost-reduction techniques for mutation testing, including HOM, can be found in [29]. Moreover, Lima and Vergilio [30] presented a mapping study on search-based HOM testing.

Jia et al. [5] introduced subsuming HOMs, which are more subtle and harder-to-kill mutants than the FOMs from which they are constructed. Subsuming HOMs can replace their constituent FOMs without loss of test effectiveness. They used search-based optimization techniques to manage the large number of possible fault combinations involved in searching for subtle HOMs. To search for HOMs, they used the Greedy Search, Hill Climbing and Genetic algorithms. As the set of test cases that can kill the HOM would also kill all the FOMs from which it is constructed, we can replace all the FOMs by the HOM created. HOM testing can reduce test effort by avoiding dumb mutants in favour of subtle ones. Fewer (but better) mutants means fewer (but better) test cases.

Polo et al. [31] presented a technique to reduce the cost of mutation testing and minimizing the percentage of equivalent mutants by using second-order mutants (SOM). Their approach combines generated first-order mutants, By doing this, the number of mutants is reduced by half. Moreover, when equivalent FOM is combined with non-equivalent FOM, the result is a non-equivalent HOM. This way, the number of equivalent mutants can also be reduced. In order to generate SOMs they used the LastToFirst, DifferentOperators and RandomMix algorithms. The LastToFirst algorithm builds SOMs by combining the first mutant with the last one, and then the second one with the previous to the last, etc. DifferentOperators combines mutants proceeding from different mutation operators. RandomMix randomly combines any two FOMs by using each mutant once. The number of mutants can be reduced by half the original suite, which reduces the effort and cost of mutation testing. Moreover, the number of equivalent mutants was reduced from 18.66% to about 5%.

Langdon et al. [20] proposed an approach by using genetic algorithms to search for higher-order mutants which are both hard to kill and syntactically similar to the original

program under test. They used the Pareto optimal algorithm [32] and explored the relationship between source code changes introduced by mutations and corresponding changes to the program's behaviour. The Pareto optimal approach in this case means preferring mutants that pass more of the test cases and are closer to the original program. To generate HOMs, they used Deb's Sorting Genetic Algorithm II [33]. In each generation, crossover is used to create a new population. The next generation is given by a Pareto optimal selection. and so on. The results showed that the genetic programming is able to find complex non-equivalent HOMs of real programs which are harder to kill than any FOM.

Omar et al. [21] proposed four approaches to construct HOMs for Java and AspectJ programs. The approaches are based on an aspect-oriented fault model given in [24], which classifies faults into faults that can occur in base classes, in aspects (pointcut, inter-type declarations, aspect declaration, and advice), or in the interaction between the base classes and aspects. They also developed a prototype tool that automates the process of constructing, compiling, and executing FOMs and HOMs against test sets. The first approach is Single Base Class or Aspect Approach. In this approach, each HOM is constructed by inserting two or more mutation faults into a single base class or a single aspect. The second approach is Dispersed Base Class Approach, wherein each HOM is constructed by inserting two or more mutation faults in two or more different base classes. The third approach is the Dispersed Aspect Approach, in which each HOM is constructed by inserting two or more mutation faults in two or more different aspects. The fourth approach is Dispersed Base Class and Aspect Approach, in which each HOM is constructed by inserting at least one fault in a base class and at least one fault in an aspect. The results showed that all approaches can produce HOMs that can be used to increase test effectiveness and reduce test effort. However, the Single Base Class or Aspect Approach produced a higher percentage of harder-to-kill HOMs and more effectively reduced the total number of the mutants.

Omar et al. [23] also proposed three algorithms: Genetic Algorithm, Local Search, and Random Search, for finding subtle HOMs for a given Java or AspectJ program. They define harder-to-kill subtle HOMs as those that are not killed by an existing test set that kills all the first-order mutants of a given program. This can be used to improve the fault detection effectiveness of test sets. To generate higher-order mutants, they first generate a population of FOMs by using their tool that uses the same approach and implementation described in their previous work [21], which utilizes MuJava [34] to create base class FOMs and AjMutator [35] to create point-cut FOMs. The generated FOMs are formulated into an XML file, which is called FOM metadata. Then two or more of the generated FOMs are combined to generate HOM in each generation by using one of the algorithms specified. All three algorithms found subtle HOMs. But Random Search found HOMs of lower (second or third) degrees, and Genetic Algorithm found subtle HOMs of higher degrees. However, Local Search was the most successful overall in finding them.

Mateo et al. [22] presented an approach to reduce the cost of mutation testing by reducing the number of FOMs when creating SOMs. They applied mutation testing on the system level instead of the unit level, because on the unit level, the tester's attention would be focused on units, and testing the behavior of other system characteristics would be beyond the scope of the process. To generate mutants, they used four algorithms to combine faults in order to obtain SOMs. The algorithms used are FirstToLast, Between-Operatos, Random and Each Choice. As the generation is done at system level, they introduced a combination restriction to all the algorithms in order to disperse the faults throughout the system. This restriction means that the combined errors must be inserted into different classes off the system. They also implemented three greedy algorithms in order to reduce a test suite based on the mutants killed by the set. Results showed that the mutation score of the FOM was around 80%, while the SOM was around 95%.

Mateo et al. [22] did an empirical study on different strategies to compose second-order mutants at system level as well as a cost–risk analysis of higher-order mutation at system level. The focus was on cost-reduction approach by reducing the mutants set through the combination of the first-order mutants into higher-order mutants. The aim of

the study was to measure the effect of second-order and first-order mutations on mutation scores. They conducted an experiment by using three reduction algorithms (MAX, MIN, and RDM), and the results have shown that, taking into account the high cost savings of second-order mutation as well as the novel test design strategy supported by mutation at system level, a second-order mutation is sufficiently effective and requires fewer resources.

Omar et al. [36] proposed new search techniques for finding subtle HOMs with new heuristics and search strategies. To generate HOMs they added Guided Local Search, which uses the same steps as Local Search, but utilizes program structural information to help it focus on the FOM combinations that are more likely to produce subtle HOMs. Restricted Random Search uses the same steps as Random Search, but as in their previous study Random Search found mutants that were of the lower degree; thus, Restricted Random Search uses a configurable parameter to allow it to control the maximum degree of generated HOMs. Restricted Enumeration Search examines candidate HOMs in the search space in a predefined sequence until a defined stopping condition is met. Guided Local Search was the most successful in terms of finding the highest number of subtle HOMs for most programs that were used. Restricted Enumeration Search was the most successful for programs that have a high number of subtle second-order mutants.

Omar et al. [37] extended their prior work and improved the search techniques they used for finding subtle HOMs. They developed six search-based techniques: Genetic Algorithm, Local Search, Data-Interaction Guided Local Search, Test-Case Guided Local Search, Restricted Random Search and Restricted Enumeration. They defined a different fitness function that equaled the summation of Difficulty of killing HOM and Fault detection difference between HOM and its constituent FOMs; the reason for including FOMs in the fitness function was the fitness measure defined by Jia and Harman [5], as it favored HOMs that are killed by fewer test cases than their constituent FOMs. The results showed that Local Search and both the Guided Local Search techniques were more effective than the other techniques at finding subtle HOMs.

Nguyen et al. [38] presented an approach to reduce the cost of mutation testing by reducing the number of HOMs, to find harder-to-kill mutants, and to reduce the cost of mutation testing by not wasting resources for creating easy-to-kill mutants. They also performed an experimental evaluation of the effect of applying optimization algorithms in the mutation testing, based on their proposed HOMs classification, objectives and fitness functions by using a tool called Judy that they originally presented. The results showed that their approach can generate harder-to-kill and more realistic HOMs and reduce the total number of generated HOMs. As a result, the approach can be used to improve the mutation testing effectiveness in general.

Abuljadayel and Wedyan [39] proposed an approach and a tool for finding subtle HOMs by using a genetic algorithm. Their approach used an enhanced mode of crossover called crossover by replacement, wherein faults from two parent mutants are replaced to produce two new mutants that have some properties of their parents. Their approach was able to generate subtle HOMs and a low percentage of equivalent mutants. The proposed approach in this paper uses a different technique for crossover and uses five selection strategies as well.

Nguyen [40] proposed a method to use higher-order mutants for creating mutants, for example, by using two second-order mutants to construct a fourth-order mutant. The results show a reduction in the number of generated mutants.

Jang et al. [41] proposed a higher-order mutation-based fault localization technique called HOTFUZ. The approach uses HOMs to reduce the cost while minimizing the accuracy degradation. The authors performed an experimental study by using 65 real-world faults. The results show that HOTFUZ outperforms alternative strategies by localizing faults more accurately by using the same number of mutants executed. Wang et al. [42] used another approach for fault localization by using HOMs. The authors performed experiments on two real-world benchmarks and found that second- and third-order mutants can help improve the fault localization performance.

## 3. Proposed Approach

Genetic algorithms (GAs), since their introduction by John Holland [43], have been widely used for solving optimization problems. Genetic algorithms are inspired by the process of natural evolution by using selection, crossover, and mutation to generate solutions for optimization problems.

The GA starts with a randomly generated population of candidate solutions called individuals. Each individual holds properties called chromosomes that go through crossover and mutation. The population then evolved toward better solutions. The initial population size depends on the problem, but it usually contains hundreds of possible solutions.

During each generation, a subset of the population is selected to produce a new generation. The selection process is based on the fitness of the individuals, where fitter solutions are more likely to be selected. To measure the fitness of individuals, a fitness function is defined.

In the following, we describe the details of the proposed genetic algorithm for solving the problem of constructing subtle mutants (GaSubtle). We start by defining the fitness function used by GaSublte, and then we proceed by describing each step of the algorithm.

### 3.1. Fitness Function

The fitness or objective function is used to evaluate a given set of candidate solutions to determine their quality. For each generation, the fitness of all individuals should be calculated to decide which ones will be dropped off or survive to the next generation.

In GaSubtle, we used the fitness function originally proposed by Omar et al. [23]. It uses information about the sets of test cases that kill the HOM and those that kill its constituent FOMs. The notations used by the function are as follows:

- $F = \{f_1, \ldots, f_n\}$ is the set of all non-equivalent FOMs for the program under test.
- $H$ is the space of all candidate HOMs. $H = \mathcal{P}(F)$, where $\mathcal{P}$ is a power set.
- $U$ is the universe of all possible test cases.
- $T = \{t_{c_1}, \ldots, t_{c_m}\}$ is the set of all test cases under consideration (the given test suite), $T \subset U$ and $T$ kills all the FOMs in $F$.
- $h_i^n \in H$ is an HOM constructed from $n$ different FOMs, such that $h_i = \{f_{i_1}, \ldots, f_{i_n}\}$. The notation can be simplified to $h_i = h_i^n$ without confusion.
- Let $T_{h_i} \subseteq T$ denote the set of those test cases in $T$ that kill $h_i$. $T_{h_i} = \phi$ if none of the test cases in $T$ kill $h_i$.
- There are $n$ test sets $T_{i_1}, \ldots, T_{i_n}, \forall j \in [1, \ldots, n], T_{i_j} \subseteq T$ and $T_{i_j}$ contains all test cases that kill $f_{i_j}$ in $h_i$.
- $TU_i$ is a test set such that $TU_i = \bigcup_{j=1}^{n} T_{i_j}$.

The fitness function combines two measures: (1) fault detection difference between the HOM and its constituent FOMs as shown in Equation (1), and (2) the difficulty of killing HOM as shown in Equation (2). The two measures are combined as shown in Equation (3). The value of $\alpha$, which lies between 0 and 1, determines the weight of the two terms.

$$FDD(h_i) = \frac{|\,(TU_i \cup T_{h_i})\,| - |\,(TU_i \cap T_{h_i})\,|}{|\,TU_i \cup T_{h_i}\,|} \tag{1}$$

$$DOK(h_i) = \frac{|\,(TU_i \cup T_{h_i})\,| - |\,T_{h_i}\,|}{|\,TU_i \cup T_{h_i}\,|} \tag{2}$$

$$fitness(h_i) = \alpha \times DOK(h_i) + (1 - \alpha) \times FDD(h_i) \tag{3}$$

The goal of the FDD term is to capture the level of interaction between the constituent FOMs of a HOM in terms of the difference between the set of test cases that kill the HOM and the union of all sets of test cases that kill each individual constituent FOM. Accordingly, a HOM is subtle when the constituent FOMs interact to mask each other and produce new faulty behavior that cannot be detected by the given test set. Therefore, when the difference

is higher, the HOM is given a higher ftiness value. The value of the FDD measure lies between 0 and 1, where 1 indicates the fittest HOM that is killed by a totally different set of test cases than its constituent FOMs.

The DOK term measures difficulty in terms of the size of the test set needed to kill the mutant. The value of DOK lies between 0 and 1, where 1 means the mutant is not killed by any test case, and therefore, has the highest fitness value.

### 3.2. Initialization

The genetic algorithm requires a set of candidate solutions, which constitute the first or initial generation.The candidate solutions in our case are HOMs (each solution represents one HOM). In GaSubtle, we randomly generated candidate mutants distributed as follows:

- First-order mutants: 10% of the generation size.
- Second-order mutants: 80% of the generation size.
- Third-order mutants: 5% of the generation size.
- Fourth-order mutants: 5% of the generation size.

The decision of having a majority of second-order mutants in the first generation is based on results of previous studies which showed that most subtle mutants are in the lower orders [25,37]. Therefore, starting with a large percentage of second-order mutants can speed up the genetic algorithm. FOMs are generated by using all mutation operators implemented in muJava [34].

Instead of setting the generation size randomly, we computed the size of the generation by using the formula given in Equation (4). Our motivation here is that the number of mutants that can be generated for any program depends on program size (along with other factors); therefore, using a generation size that depends on the program size can speed up the GA in finding subtle mutants:

$$generationsize = \beta \times LOC \tag{4}$$

where $\beta$ = any double value from 0.5 to 3.0 and $LOC$ = lines of source code, ignoring comments and empty lines.

### 3.3. Selection

At each generation, a set of individuals are selected out of the population in order to perform mutation and crossover. The selection process is based on the fitness of the individuals, where fitter solutions are more likely to be selected. Many selection strategies are available; some of these strategies measure the fitness of each individual and select the fittest solutions. Other strategies measure only a sample of the population as computing the fitness function, but these can be time-consuming.

In GaSubtle, we used five selection strategies. These are:

1.  Roulette Wheel Selection [44]: Selects $n$ random candidates, where the probability of each candidate getting selected is proportional to its fitness score. Candidates may get selected more than once. In some cases, particularly with small population sizes, the randomness of selection may result in excessively high occurrences of particular candidates.
2.  Stochastic Universal Sampling [45]: An alternative to Roulette Wheel Selection as a fitness-proportionate selection strategy. It ensures that the frequency of selection for each candidate is consistent with its expected frequency of selection.
3.  Tournament Selection [46]: Selects a random pair of candidates and then selects the fitter of the two candidates with probability $p$, where $p$ is the configured selection probability therefore the probability of the least fit candidate being selected is 1—$p$.
4.  Truncation Selection [47]: Selects $n$ candidates from a population by simply selecting $n$ candidates with the highest fitness value (the rest are discarded). A candidate is never selected more than once.
5.  Random Selection [48]: Selects candidates from a population at absolute randomness.

### 3.4. Mutation

The purpose of the mutation process in genetic algorithms is to prevent premature convergence. That is, most of the genes in the solutions share the same value before the genetic algorithm converges to a satisfactory solution [49].

GaSubtle applies mutation to a HOM by adding or removing a FOM. The decision of adding or removing an FOM is made randomly. For example, a third-order mutant might become a fourth-order or a second-order mutant after mutation. The exceptions are second-order mutants, for which we only add a FOM so that the mutation process will not produce a FOM [44].

### 3.5. Crossover

GaSubtle performs crossover between two HOMs by executing the following steps:

- Find the fitness of each FOM in the participating HOMs
- Decide the order of the generated children. The order of the generated children will be the average order of the parents. Both children will have the same order if the sum of the parents' order is even. If the sum is odd, one child will have a higher order (higher by one). For example, if the parent HOMs are of orders two and four, then the two children will have an order of three. If the parent HOMs are of orders two and five, then one child will be of order three and the other will be of order four.
- Select the fittest FOMs from both parents and place them in the first child, and the least fit FOMs in the second child. If the two parents differ in order (e.g., second-order mutant crossover with third-order mutant) then the fittest child will be in the lower order whereas the least fit child will have an order higher by one.

We illustrate the selection process abstractly in the example given in Figure 1. In the figure, we represent each parent solution with a box and the number and fitness of each FOM it has. Parent *A* is a third-order mutant built from FOMs with fitness values of 0.5, 0.9, and 0.6. When parent *A* is selected for crossover with parent **B**, which is a second-order mutant, the two produced children will be of orders two and three. Child A is produced by using the least fit three FOMs from both parents, whereas child B is produced by using the fittest two FOMs from each parent.

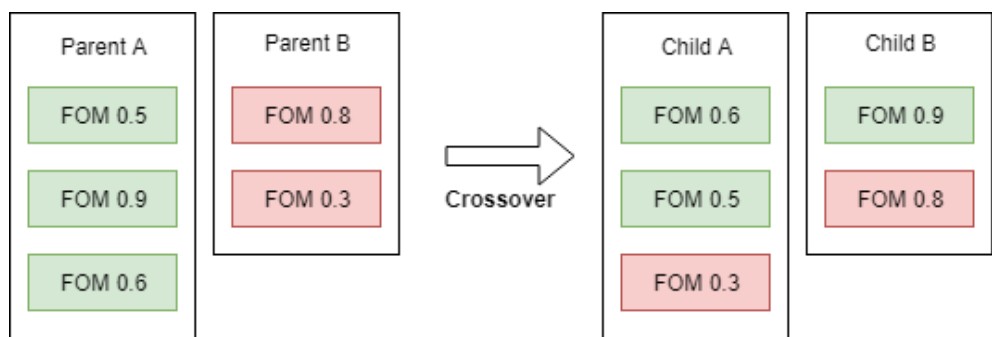

**Figure 1.** Proposed crossover example.

The following example illustrates how the proposed crossover works by using java source code. The original program, called PatternIndex (https://cs.gmu.edu/~offutt/softwaretest/java/PatternIndex.java accessed on 1 May 2022). and shown in Figure 2, searches for a given pattern in a given string and returns the beginning index of that pattern. In Figure 3, we show the first parent selected for crossover. The first parent is a third-order mutant, built from FOM 1.1 with fitness 0.3, FOM 1.2 with fitness 0.5, and FOM 1.3, with fitness 0.1. The FOMs are shown in a box in the code with a comment that displays the mutant name and fitness. Similarly, Figure 4 shows the other parent selected for crossover, which is a second-order mutant built from FOM 2.1 with fitness value of 0.4, and FOM 2.2 with fitness value of 0.2.

```
public class PatternIndex {
 public static int patternIndex(String subject, String pattern) {
  final int NOTFOUND = -1;
  int iSub = 0;
  int rtnIndex = NOTFOUND;
  boolean isPat = false;
  int subjectLen = subject.length();
  int patternLen = pattern.length();
  while (isPat == false && iSub + patternLen - 1 < subjectLen) {
   if (subject.charAt(iSub) == pattern.charAt(0)) {
     rtnIndex = iSub;
     isPat = true;
     for (int iPat = 1; iPat < patternLen; iPat++) {
      if (subject.charAt(iSub + iPat) != pattern.charAt(iPat)) {
         rtnIndex = NOTFOUND;
         isPat = false;
         break;
      }
     }
   }
  iSub++;
 }
 return rtnIndex;
 }
}
```

**Figure 2.** Original program.

```
public class PatternIndex {
 public static int patternIndex(String subject, String pattern) {
  final int NOTFOUND = -1;
  int iSub = 0;
  int rtnIndex = NOTFOUND;
  boolean isPat = false;
  int subjectLen = subject.length() - 1;    //FOM 1.1, fitness = 0.3
  int patternLen = pattern.length();
  while (isPat == false && iSub + patternLen - 1 < subjectLen) {
   // FOM 1.2, fitness = 0.5
   if (subject.charAt(iSub) == pattern.charAt(1)) {
       rtnIndex = iSub;
       isPat = false;   // FOM 1.3 - fitness = 0.1
       for (int iPat = 1; iPat < patternLen; iPat++) {
        if (subject.charAt(iSub + iPat) != pattern.charAt(iPat)) {
          rtnIndex = NOTFOUND;
          isPat = false;    // FOM 1.3 - fitness = 0.1
          break;
       }
      }
   }
  iSub++;
 }
 return rtnIndex;
 }
}
```

**Figure 3.** Parent 1: Third-order mutant selected for crossover.

```
public class PatternIndex {
 public static int patternIndex(String subject, String pattern){
  final int NOTFOUND = -1;
  int iSub = 1;   // FOM 2.1 - fitness = 0.4
  int rtnIndex = NOTFOUND;
  boolean isPat = true;   // FOM 2.2 - fitness = 0.2
  int subjectLen = subject.length();
  int patternLen = pattern.length();
  while (isPat == false && iSub + patternLen - 1 < subjectLen){
   if (subject.charAt(iSub) == pattern.charAt(0)) {
       rtnIndex = iSub;
       isPat = true;
       for (int iPat = 1; iPat < patternLen; iPat++) {
         if (subject.charAt(iSub + iPat) != pattern.charAt(iPat)) {
          rtnIndex = NOTFOUND;
          isPat = false;
          break;          }
       }
   }
   iSub++;
  }
  return rtnIndex;
 }
}
```

**Figure 4.** Parent 2: Second-order mutant selected for crossover.

When crossover is performed, the least fit three FOMs from both parents are used to create the first child, and the most fit two FOMs from both parents are used to create the second child. Therefore, child one is built by using FOM 1.1. FOM 1.3, and FOM 2.2. Child 1 is shown in Figure 5. Child 2 is built by using FOM 1.2, and FOM 2.1, as shown in Figure 6.

```
public class PatternIndex {
 public static int patternIndex(String subject, String pattern){
  final int NOTFOUND = -1;
  int iSub = 0;
  int rtnIndex = NOTFOUND;
  boolean isPat = true;   // FOM 2.2 - fitness = 0.2
  int subjectLen = subject.length() - 1;   //FOM 1.1, fitness = 0.3
  int patternLen = pattern.length();
  while (isPat == false && iSub + patternLen - 1 < subjectLen){
   if (subject.charAt(iSub) == pattern.charAt(0)) {
       rtnIndex = iSub;
       isPat = false;   // FOM 1.3 - fitness = 0.1
       for (int iPat = 1; iPat < patternLen; iPat++) {
         if (subject.charAt(iSub + iPat) != pattern.charAt(iPat)) {
          rtnIndex = NOTFOUND;
          isPat = false;
          break;          }
       }
   }
   iSub++;
  }
  return rtnIndex;
 }
}
```

**Figure 5.** Child 1: Third-order mutant constructed by using the least fit FOMs from parent 1 and parent 2.

```java
public class PatternIndex {
 public static int patternIndex(String subject, String pattern){
  final int NOTFOUND = -1;
  int iSub = 1;  // FOM 2.1 - fitness = 0.4
  int rtnIndex = NOTFOUND;
  boolean isPat = false;
  int subjectLen = subject.length();
  int patternLen = pattern.length();
  while (isPat == false && iSub + patternLen - 1 < subjectLen){
   // FOM 1.2, fitness = 0.5
   if (subject.charAt(iSub) == pattern.charAt(1)) {
   rtnIndex = iSub;
   isPat = true;
   for (int iPat = 1; iPat < patternLen; iPat++) {
        if (subject.charAt(iSub + iPat) != pattern.charAt(iPat)) {
        rtnIndex = NOTFOUND;
        isPat = false;
        break;}
   }
  }
  iSub++;
 }
 return rtnIndex;
 }
}
```

**Figure 6.** Child 2: Second-order mutant, constructed by using the most fit FOMs from parent 1 and parent 2.

*3.6. Termination*

The genetic algorithm keeps iterating selection, crossover, and mutation until a termination condition is met. There are various alternative stopping conditions for a genetic algorithm. GaSubtle can be terminated when one of the following conditions is satisfied:

- Reaching a given number of overall generated HOMs.
- Reaching a given number of subtle HOMs are found.
- Reaching a given number of generations.
- Timeout.

The termination condition is passed as an option to the algorithm. The pseudocode for GaSubtle is given in Algorithm 1.

---

**Algorithm 1** Genetic Algorithm

---

**Require:** FOMsList, mutationRate, maxDegree
**Ensure:** maxDegree > 1
 1: liveMutants ← $\phi$
 2: subtleMutants ← $\phi$
 3: population ← generateRandomPopulation()
 4: execute(population)
 5: evaluate(population)
 6: **while** !stopConditionMet() **do**
 7:　newMutants ← select(population)
 8:　crossover(newMutants)
 9:　mutate(newMutants)
10:　execute(newMutants)
11:　evaluate(population)
12:　population.removeLeastFitMutants()
13:　population.addMutants(newMutants)
14:　liveMutants ← population.getLiveMutants()
15:　subtleMutants ← population.getSubtleMutants()
16: **end while**
17: **return** subtleMutants

---

## 4. Gasubtle Tool

In this section, we overview the design and implementation of the tool we implemented for finding subtle mutants. The tool, which we named GaSubtle (https://github.com/AbdullahAsendar/GaSubtle, accessed on 1 May 2022), after the approach, automates the generation, compilation, and execution of FOMs and HOMs for Java programs.

GaSubtle is built by using a layered architecture pattern in order to support portability. The layers of GaSubtle are shown in figure in Figure 7. In the following sections, we describe each of the tool layers.

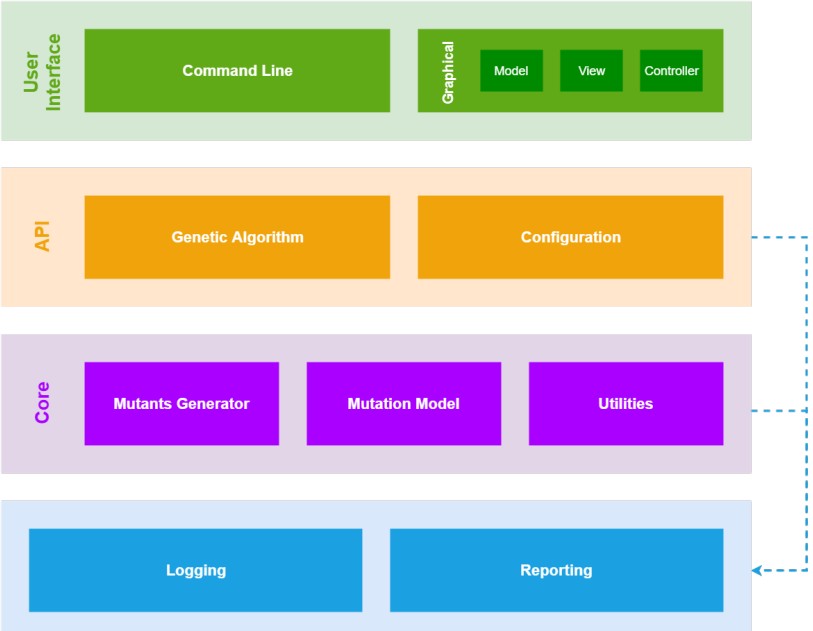

**Figure 7.** GaSubtle architecture.

### 4.1. User Interface

The user interface layer contains both graphical user interface (GUI) and a command line interface. Both interfaces are separated from the underlying layers, which allow for the modifying or adding of other user interfaces in the future without affecting existing code.

The command line interface is based on Apache Commons CLI (https://commons.apache.org/proper/commons-io, accessed on 1 May 2022), which provides an API for parsing command line options passed to the tool. GaSubtle can be run by executing java -jar GaSubtle-x.x.x.jar where GaSubtle-x.x.x.jar is the tool jar file. The command line interface takes options from Virtual Machine Arguments and properties file.

The GUI of the tool is built by using the model view controller (MVC) design pattern based on JavaFX (https://www.oracle.com/technetwork/java/javafx/overview/index.html, accessed on 1 May 2022) and Spring Framework (https://spring.io, accessed on 1 May 2022). The GUI provides an interface for passing options and displaying results.

### 4.2. Api: Application Programming Interface.

The API layer contains the genetic algorithm implementation and configurations used for tuning the algorithm. In implementing the GA, we used a Message Listener that keeps the upper layers updated with what the algorithm is doing. Therefore, the user interface layer can log these messages or show them in a graphical component.

### 4.3. Core

The core layer contains all the components needed to generate, compile, store, and execute mutants.

### 4.3.1. Mutants Generator

This component generates FOMs by using MuJava [34], which is a well-known research tool for generating FOM and performing mutation analysis. GaSubtle uses MuJava correspondent modules for generating FOM by using method-level mutation operators, and class-level mutation operators.

### 4.3.2. Mutation Model

This component stores and manages the FOMs generated by the Mutants Generator and generates HOMs. The HOMs are generated randomly by combining FOMs together. FOMs must have the seeded faults in different statements, meaning that a FOM with a seeded fault at statement 5 can not be combined with a FOM with a fault seeded at the same statement. That is because these faults may override each other and will most likely lead to an uncompilable mutant. Table 1 shows examples of compilable and uncompilable HOMs.

### 4.3.3. Utilities

This component provides the tool with a set of utilities that help in generating, compiling, and executing the HOMs. These include

- Command Line Utilities: Contains operations that utilizes executing command line commands.
- Compiler: There were many compiler implementations available. However, most of these compilers were too heavy and need a lot of time to compile mutants. This was a critical issue because the generated HOMs cannot all be compiled at once because they all represent the same process. Thus, the compiler of GaSubtle uses Spoon API [50], which is an open-source library that gives the ability of transforming and analyzing Java source code.
- Test Executor: Test execution is performed using the Command Line Utilities.

**Table 1.** Compilable and uncompilable HOMs example.

| Program | Source Code |
| --- | --- |
| Original | int sum(int n1, int n2) {<br>int sum = n1 + n2;<br>return sum;<br>} |
| Mutant 1 | int sum(int n1, int n2) {<br>int sum = ++n1 + n2;<br>return sum;<br>} |
| Mutant 2 | int sum(int n1, int n2) {<br>int sum = n1++ + n2;<br>return sum;<br>} |
| Mutant 3 | int sum(int n1, int n2) {<br>int sum = n1 + n2;<br>return ++sum;<br>} |
| Uncompilable HOM<br>(by combining mutants 1 & 2) | int sum(int n1, int n2) {<br>int sum = ++n1++ + n2;<br>return sum;<br>} |
| Compilable HOM<br>(by combining mutants 1 & 3) | int sum(int n1, int n2) {<br>int sum = ++n1 + n2;<br>return ++sum;<br>} |

## 5. Empirical Evaluation

Here, we present the details of the empirical study we performed in order to answer our research questions. Section 5.1 describes the subject programs used in the study. Section 5.2 states the research questions. Section 5.3 describes the tools used to conduct the study and Section 5.4 describes the experiment setup and execution.

### 5.1. Subject Programs

We used 10 Java programs in the experiment. The selected programs contain various constructs and solve different problems. Table 2 gives the details of the subject programs including lines of code (LoC) in column 2, number of FOM generated for the program in column 3, and the number of test cases to kill these mutants in column 4.

**Table 2.** Subject programs.

| Subject Program | LOC | No. FOMs | No. Test Cases |
| --- | --- | --- | --- |
| BinarySearch | 33 | 175 | 62 |
| Charge | 29 | 132 | 75 |
| Complex | 55 | 239 | 87 |
| Counter | 38 | 62 | 67 |
| Euclid | 26 | 106 | 95 |
| Gaussian | 58 | 457 | 90 |
| Harmonic | 15 | 57 | 72 |
| LongestCommonSubsequence | 41 | 371 | 78 |
| ArrayList | 130 | 380 | 105 |
| PatternIndex | 49 | 175 | 90 |

In the following, we briefly describe these programs:

1.  Binary search: The program implements the binary search algorithm that finds the position of a target value within a sorted array. It takes as an input a string key to search for and a sorted string array to search in. It then recursively calls a methods that performs binary search and returns the index of the provided key.
2.  Charge https://introcs.cs.princeton.edu/java/32class/Charge.java.html (accessed on 1 May 2022): This is a data type to define charged particles. It is based on Coulomb's law which says that the electric potential at a point $(x, y)$ due to a given charged particle is $V = kq/r$, where $q$ is the charge value, $r$ is the distance from the point $(x, y)$ to the charge, and $k = 8.99 \times 10^9$ is the electrostatic constant.
3.  Complex https://introcs.cs.princeton.edu/java/32class/Complex.java.html (accessed on 1 May 2022): This is a data type used to represent a complex number. A complex number is a number of the form $x + iy$, where $x$ and $y$ are real numbers and $i$ is the square root of $-1$. The basic operations on complex numbers are to add and multiply them.
4.  Counter https://introcs.cs.princeton.edu/java/33design/Counter.java.html (accessed on 1 May 2022): The program is used for counting. It encapsulates a single integer and ensures that the only operation that can be performed on the integer is increment by one.
5.  Euclid https://introcs.cs.princeton.edu/java/23recursion/Euclid.java.html (accessed on 1 May 2022): This is an implementation of the Euclidean algorithm, which is an algorithm for finding the greatest common divisor of two numbers.
6.  Gaussian https://introcs.cs.princeton.edu/java/21function/Gaussian.java.html (accessed on 1 May 2022): This implements some of the normal distribution functions, which is characterized by the familiar bell-shaped curve.

7.  Harmonic https://introcs.cs.princeton.edu/java/21function/Harmonic.java.html (accessed on 1 May 2022): This calculates the harmonic of a given number, which is the sum of the reciprocals of the first n natural numbers as shown in Equation (5).

$$H_n = 1 + \frac{1}{2} + \frac{1}{3} + \cdots + \frac{1}{n} = \sum_{k=1}^{n} \frac{1}{k} \tag{5}$$

8.  LongestCommonSubsequence https://introcs.cs.princeton.edu/java/23recursion/LongestCommonSubsequence.java.html (accessed on 1 May 2022): This implements the longest common sub-sequence problem, which is the problem of finding the longest sub-sequence common to all sequences in a given set of sequences.
9.  ArrayList https://docs.oracle.com/javase/8/docs/api/java/util/ArrayList.html (accessed on 1 May 2022): The program is a simplified implementation of Java Collections ArrayList.
10. PatternIndex https://cs.gmu.edu/~offutt/softwaretest/java/PatternIndex.java (accessed on 1 May 2022): The program searches for a given pattern in a given string and returns the beginning index of that pattern.

*5.2. Research Questions*

The experiment aims at answering the following research questions:

**RQ1**: Which selection strategy generates the highest number of subtle mutants?
**RQ2**: Does the proposed crossover generate more subtle mutants, compared with single-point crossover?
**RQ3**: Which selection strategy generates the least percentage of equivalent mutants?
**RQ4**: Does the proposed crossover generate a lesser percentage of equivalent mutants, compared with single-point crossover?
**RQ5**: What is the percentage of the generated HOMs from each mutation order?
**RQ6**: Which selection strategy has the least execution cost?
**RQ7**: Which crossover technique has the least execution cost?

*5.3. Used Tools*

The tools used to conduct this experiment are categorized into the following.

### 5.3.1. Mutation Testing Tools

A mutation testing tool automates the process of generating, compiling, and executing mutants. There are many research mutation tools available. However, in this experiment the following tools were used:

*   MuJava [34]: This is a Java-based mutation testing tool developed through the collaboration between the Korean Advanced Institute of Science and Technology in South Korea and George Mason University in the USA. MuJava is widely used research for performing mutation analysis. In this experiment, we used MuJava to generate FOMs.
*   GaSubtle: A tool we developed to implement the proposed approach for constructing subtle mutants.

### 5.3.2. Test Cases Generation Tools

A test cases generation tool automates the process of generating and organizing test cases. There are many open-source and commercial test-generation tools available. In this experiment the following tools are used:

*   Randoop [51]: This is an open source unit test generator for Java. It automatically creates unit tests for the provided classes, in JUnit format.
    Randoop generates unit tests by using feedback-directed random test generation. It is done by executing the sequences it creates, using the results of the execution to create assertions that capture the behavior of the provided classes [51].

- EvoSuite [52]: This is an open-source unit test generator for Java. It uses search-based approach integrating techniques such as hybrid search, dynamic symbolic execution, and testability transformation in order to generate JUnit test cases for a provided Java class.
- Parasoft Jtest https://www.parasoft.com/products/jtest (accessed on 1 May 2022): A commercial tool by Parasoft which provides a set of tools such as static analysis and security, unit testing for active development, unit testing for legacy code, coverage analysis and traceability, and reporting.

### 5.3.3. Test Coverage Tools

Test coverage tools automates the evaluation of test suites according to their ability to satisfy test coverage criteria. In this experiment, we used EclEmma https://www.eclemma.org (accessed on 1 May 2022), which is an open-source Eclipse IDE plug-in that computes coverage for Java programs.

### *5.4. Experiment Setup*

This section presents the setup of the experiment conducted in this paper. First, the mutants generation approach is presented. Then the test case generation approach is also presented. Finally, the configurations and the execution approach is described.

### 5.4.1. Mutants Generation

The first step was to generate FOMs for the subject programs. Table 2 shows the number of generated FOMs for the subject programs. The FOMs are used to evaluate generated test cases and are also the inputs for GaSubtle. The FOMs were generated by using MuJava [34].

### 5.4.2. Finding Test Suites

We used the test generation tools (*Randoop* [51], *EvoSuite* [52] and *Jtest*) to generate a large pool of test cases for each subject program. The pool satisfies the following two criteria: (1) it achieves full branch coverage for the subject program, and (2) it kills all FOMs for the subject program (excluding equivalent mutants). If all the generated test cases did not satisfy the two criteria, then test cases are added manually.

A test suite for a subject program is generated from the pool by repeatedly adding test cases to a suite until the desired full coverage is obtained and all FOM are killed. In each iteration, a new test case is picked up from the pool and added to the test suite. If the test case increases the suite branch coverage, or kills a live mutant, the test case is kept. Otherwise, the test case is discarded.

Table 2 shows the size of the test suite that achieve full coverage and kills all mutants for each of the subject programs.

### 5.4.3. Configuration

The experiment was conducted on a PC running Ubuntu 18.04 with a fourth-generation Intel Core i7 processor and 16 GB of RAM. The genetic algorithm was set up as follows:

- The value of $\alpha$ of the fitness function was set at 0.75.
- Termination condition was set to reach 300 generations.
- Maximum mutation order not to be exceeded was set to 5.
- Mutation percentage was set to 5%.

For each subject program, the experiment was executed 10 times with each selection strategy.

## 6. Results and Analysis

In this section, we provide answers for the research questions and present our findings. In the following sections, the results given are computed by taking the average of 10 experiments performed on each of the subject programs.

*6.1. Rq1: Which Selection Strategy Generates the Highest Number of Subtle Mutants?*

To answer this question, we generated HOMs by using single-point crossover, and the proposed crossover with each selection strategy.

6.1.1. Single-Point Crossover

The results of the experiment for single-point crossover are shown in Table 3. Columns 2 through 6 show the average number of generated subtle mutants using each selection strategy for each subject program. In each row, the selection strategy with the highest subtle mutants is given in bold font. Truncation and stochastic universal sampling selection strategies each generated the highest number of subtle mutants in three subject programs. However, in total, truncation selection strategy generated more subtle mutants. Roulette Wheel Selection generated the minimum number of subtle mutants in total and generated the highest number of subtle mutants in only one program.

**Table 3.** Number of subtle mutants generated using single point crossover.

| Subject Program | Selection Strategy | | | | |
| --- | --- | --- | --- | --- | --- |
| | Random | Roulette Wheel | SUS | Tournament | Truncation |
| BinarySearch | 1676 | 1141 | **1935** | 1264 | 877 |
| Charge | 1008 | 160 | 383 | **1534** | 245 |
| Complex | 27 | **79** | 4 | 41 | 14 |
| Counter | 847 | 685 | **1175** | 1137 | 732 |
| Euclid | 205 | 152 | 256 | 238 | **268** |
| Gaussian | **70** | 39 | 61 | 19 | 27 |
| Harmonic | 201 | 236 | 380 | 300 | **2029** |
| LongestCommonSubsequence | 65 | 67 | **90** | 25 | 15 |
| ArrayList | 2268 | 1632 | 2587 | 1561 | **3412** |
| PatternIndex | 829 | 728 | 564 | **1018** | 521 |
| Total | 7196 | 4919 | 7435 | 7137 | **8140** |

6.1.2. Proposed Crossover

The results of the experiment for the proposed crossover are shown in Table 4. The table is organized in the same way as Table 3. Truncation selection generated the highest number of subtle mutants in four subject programs and in total. However, truncation selection did not generate any subtle mutants for two programs. This result suggests a dependency relation between the source code and the used selection strategy.

**Table 4.** Number of subtle mutants generated using proposed crossover.

| Subject Program | Selection Strategy | | | | |
| --- | --- | --- | --- | --- | --- |
| | Random | Roulette Wheel | SUS | Tournament | Truncation |
| BinarySearch | 4416 | 2498 | 3184 | 2451 | **7626** |
| Charge | 891 | **1427** | 293 | 436 | 464 |
| Complex | **276** | 235 | 59 | 56 | 0 |
| Counter | 2644 | 2421 | 2200 | **5764** | 3534 |
| Euclid | 529 | 386 | 359 | 326 | **559** |
| Gaussian | 110 | 120 | 60 | 115 | **316** |
| Harmonic | 677 | 525 | 400 | **960** | 252 |
| LongestCommonSubsequence | 29 | 107 | **126** | 40 | 0 |
| ArrayList | 5019 | 2862 | 2649 | 3511 | **6744** |
| PatternIndex | 441 | 1291 | **1068** | 777 | 994 |
| Total | 15,032 | 11,872 | 10,398 | 14,436 | **20,489** |

*6.2. Rq2: Does the Proposed Crossover Generate More Subtle Mutants Compared with Single-Point Crossover?*

Tables 3 and 4 show the total number of generated subtle mutants for both single-point crossover and proposed crossover.

For each single one of the selection strategies, the proposed crossover generated significantly more subtle mutants. Comparing the subject programs, the proposed crossover also generated more subtle mutants except for the Harmonic program. The proposed crossover generally generates more subtle mutants compared with single-point crossover.

*6.3. Rq3: Which Selection Strategy Generates the Smallest Percentage of Equivalent Mutants?*

To calculate the average equivalent mutants generated by each selection strategy for both proposed crossover and single-point crossover, a subset of generated subtle HOMs was examined. The examination was done by inspection of a random 100 mutants of the generated subtle HOMs for each selection strategy. However, some of the selection strategies generated less than 100 subtle HOMs; in such cases, all of the generated subtle HOMs were examined.

With no alive FOMs entered into the genetic algorithm, and with a test set that scored a 100% coverage, the number of equivalent mutants can be minimized. However, combining two non-equivalent FOMs can still result in an equivalent HOM.

### 6.3.1. Single-Point Crossover

Table 5 shows the percentage of generated equivalent mutants in each selection strategy for each subject program by using single-point crossover. Although all five selection strategies generated equivalent mutants, there are significant differences in the percentage of equivalent mutants generated. For the single-point crossover, truncation selection recorded the minimum percentage of 1.3%. Secondly, the tournament selection has a slightly higher percentage of 1.4%. The selection strategy to generate the maximum number of equivalent mutants was a random selection with a percentage of 6.2%.

For the subject programs, PatternIndex recorded the minimum percentage of equivalent mutants with the percentage of 3.8%, and Euclid recorded the maximum percentage of equivalent mutants across all selection strategies with the percentage of 16.2%.

**Table 5.** Percentage of equivalent HOMs generated by using single-point crossover.

| Subject Program | Selection Strategy | | | | |
| --- | --- | --- | --- | --- | --- |
| | **Random** | **Roulette Wheel** | **SUS** | **Tournament** | **Truncation** |
| BinarySearch | 8% | 5% | 9% | 3% | 6% |
| Charge | 4% | 2.7% | 8% | 3% | 5.6% |
| Complex | 9.7% | 7.4% | 0% | 7.3% | 3% |
| Counter | 10% | 9% | 7% | 8% | 8% |
| Euclid | 14% | 13.2% | 20.5% | 15.7% | 16.1% |
| Gaussian | 13.6% | 5.1% | 3.2% | 5.2% | 3.7% |
| Harmonic | 5% | 6% | 5% | 7% | 4% |
| LongestCommonSubsequence | 10.7% | 5.4% | 3.3% | 9% | 6.7% |
| ArrayList | 5% | 4% | 3% | 6% | 5% |
| PatternIndex | 2% | 4% | 5% | 5% | 3% |
| Total | 6.2% | 2.2% | 2.4 % | 1.4% | 1.3% |

### 6.3.2. Proposed Crossover

Table 6 shows the percentage of generated equivalent mutants in each selection strategy for each subject program using the proposed crossover. Unlike the single-point crossover, there were slight differences in the the percentage of equivalent mutants generated for each selection strategy. The minimum percentage was 1.4% and the maximum was 4.8%. For the proposed crossover, tournament selection recorded the minimum percentage of 1.4%. Secondly, the truncation selection had a slightly higher percentage of 1.7%. The selection strategy to generate the maximum number of equivalent mutants was, again, a random selection with a percentage of 4.8%.

For the subject programs, Gaussian recorded the minimum percentage of equivalent mutants with the percentage of 1.7%, and Euclid again recorded the maximum percentage of equivalent mutants across all selection strategies with the percentage of 7.3%.

For both single-point crossover and the proposed crossover, truncation and tournament selection generated the minimum percentage of equivalents mutants, and random selection generated the maximum percentage. The Gaussian subject program recorded the minimum percentage of equivalents mutants and Euclid recorded the maximum.

**Table 6.** Percentage of equivalent HOMs generated by using proposed crossover.

| Subject Program | Selection Strategy | | | | |
| --- | --- | --- | --- | --- | --- |
| | **Random** | **Roulette Wheel** | **SUS** | **Tournament** | **Truncation** |
| BinarySearch | 4% | 3% | 10% | 4% | 8% |
| Charge | 4% | 5% | 3% | 3% | 4% |
| Complex | 6% | 5% | 8.5% | 5.1% | 0% |
| Counter | 9% | 7% | 8% | 6% | 7% |
| Euclid | 12% | 7% | 5% | 8% | 4% |
| Gaussian | 2% | 1% | 3% | 1% | 2% |
| Harmonic | 3% | 5% | 6% | 4% | 2% |
| LongestCommonSubsequence | 3.4% | 3% | 5% | 2.5% | 0% |
| ArrayList | 3% | 4% | 6% | 5% | 4% |
| PatternIndex | 3% | 2% | 5% | 3% | 4% |
| Total | 4.8% | 1.9% | 2.1% | 1.4% | 1.7% |

### 6.4. Rq4: Does the Proposed Crossover Generate a Lesser Percentage of Equivalent Mutants Compared with Single-Point Crossover?

Tables 5 and 6 shows the percentage of both single-point crossover and proposed crossover-generated equivalent mutants.

By using random, Roulette Wheel, stochastic universal sampling, and tournament selections, the proposed crossover generated less equivalent mutants compared with single-point crossover. However, by using truncation selection, the single-point crossover generated slightly less equivalent mutants. Comparing the subject programs, the proposed crossover generated less equivalent mutants the the single-point crossover except for Harmonic and Charge programs.

The proposed crossover generally generates less equivalent mutants compared with single-point crossover. However, under certain circumstances the single-point crossover can generate less equivalent mutants.

### 6.5. Rq5: What Is the Percentage of the Generated HOMs from Each Mutation Order?

GaSubtle records the state of each generation containing the generation size and the order of the mutants in each generation.

#### 6.5.1. Single-Point Crossover

Table 7 shows the percentage of each mutation order for the generated mutants by using single-point crossover. We show the percentage over all subject programs for space limitations. Most of the mutants were of the second order for all selection strategies with the percentage of 79.5%. Then comes the third order with the percentage of 15.5%, the fourth order with 4.1%, and the fifth order with 0.8%. The gap between the percentage of second- and third-order mutants was 64%.

**Table 7.** Percentage of the generated HOMs from each mutation order by using single-point crossover.

| Mutation Order | Selection Strategy | | | | |
|---|---|---|---|---|---|
| | Random | Roulette Wheel | SUS | Tournament | Truncation |
| Second | 60.2 | 88.2 | 90.1 | 88.2 | 74.2 |
| Third | 28.5 | 7.7 | 7.7 | 8.5 | 22 |
| Fourth | 8.9 | 3.4 | 1.9 | 1.8 | 2.1 |
| Fifth | 2.5 | 0.7 | 0.4 | 0.2 | 0.4 |

6.5.2. Proposed Crossover

Table 8 shows the percentage of each mutation order for the generated mutants by using the proposed crossover. The second order had the highest percentage of 54.8%, then the third order 26.8%, fourth order 14.5% and the fifth order 3.9%. The gap between the percentage of second- and third-order mutants was 28%, which is much less than the single-point crossover.

For both single-point crossover and the proposed crossover, most of the generated mutants were in the second order. However, the proposed approach was able to find subtle mutants at higher mutation orders. Searching for subtle mutants at higher order is hard due to the size of the search space. The proposed crossover was able to find about 25% more subtle mutants at higher orders, compared with single crossover.

**Table 8.** Percentage of the generated HOMs from each mutation order by using proposed crossover.

| Mutation Order | Selection Strategy | | | | |
|---|---|---|---|---|---|
| | Random | Roulette Wheel | SUS | Tournament | Truncation |
| Second | 21.5 | 59.1 | 57.1 | 58.7 | 72.9 |
| Third | 37.1 | 25.2 | 28 | 25.9 | 20.2 |
| Fourth | 30.6 | 12.9 | 12.8 | 12.7 | 5.8 |
| Fifth | 10.8 | 2.8 | 2.1 | 2.6 | 1.2 |

*6.6. Rq6: Which Selection Strategy Has the Least Execution Cost?*

GaSubtle has a timer that starts when the algorithm starts execution and stops when the algorithm terminates. Then the execution time is exported to the generated report. In order to calculate the execution cost, we recorded the total run-time for each one of the selection strategies.

6.6.1. Single-Point Crossover

Table 9 shows the average run-time in minutes for each one of the subject programs using single-point crossover. Truncation selection recorded the minimum run-time average of 16 min. Both Roulette Wheel Selection and stochastic universal sampling recorded the maximum run-time average of 22.2 min.

6.6.2. Proposed Crossover

Tables 10 shows the average run-time in minutes for each one of the subject programs using the proposed crossover. Truncation selection again recorded the minimum run-time average of 13.3 min. Again, both Roulette Wheel Selection and stochastic universal sampling recorded the maximum run-time average of 17.3 and 17.2 min, respectively.

For both single-point crossover and the proposed crossover, truncation selection had the least execution cost.

**Table 9.** Average run time of single-point crossover in minutes.

| Subject Program | Selection Strategy | | | | |
| --- | --- | --- | --- | --- | --- |
| | Random | Roulette Wheel | SUS | Tournament | Truncation |
| BinarySearch | 17.2 | 16.1 | 17.3 | 17.2 | 11.8 |
| Charge | 14.9 | 14.8 | 15.2 | 15.1 | 14.1 |
| Complex | 25.2 | 26.7 | 28.1 | 13.1 | 11.6 |
| Counter | 16.6 | 17.3 | 17.2 | 16.8 | 12.2 |
| Euclid | 15.1 | 13.4 | 14.1 | 14.6 | 13.4 |
| Gaussian | 29.6 | 31.5 | 33.2 | 24.8 | 25.1 |
| Harmonic | 16.5 | 16.2 | 16.4 | 16.8 | 15.4 |
| LongestCommonSubsequence | 27.1 | 29.4 | 28.9 | 23.2 | 18.6 |
| ArrayList | 23.5 | 31.2 | 27.6 | 24.1 | 19.4 |
| PatternIndex | 21.3 | 25.8 | 25.2 | 20.9 | 18.5 |
| Total | 20.7 | 22.24 | 22.32 | 18.66 | 16.01 |

**Table 10.** Average run-time of proposed crossover in minutes.

| Subject Program | Selection Strategy | | | | |
| --- | --- | --- | --- | --- | --- |
| | Random | Roulette Wheel | SUS | Tournament | Truncation |
| BinarySearch | 14.3 | 9.1 | 13.9 | 14.2 | 13.6 |
| Charge | 12.8 | 13.4 | 12.7 | 13.1 | 11.3 |
| Complex | 17.9 | 28.6 | 29.3 | 17.2 | 15.3 |
| Counter | 8.3 | 8.2 | 7.9 | 8.1 | 7.8 |
| Euclid | 13.1 | 13.2 | 13.4 | 11.9 | 11.3 |
| Gaussian | 24.6 | 29.2 | 28.3 | 21.1 | 17.9 |
| Harmonic | 12.5 | 13.2 | 12.9 | 12.1 | 15.4 |
| LongestCommonSubsequence | 21.1 | 23.3 | 20.7 | 17.1 | 14.1 |
| ArrayList | 19.6 | 20.1 | 18.2 | 18.4 | 13.1 |
| PatternIndex | 15.4 | 16.3 | 15.3 | 16.2 | 13.7 |
| Total | 15.96 | 17.46 | 17.26 | 14.94 | 13.35 |

*6.7. Rq7: Which Crossover Technique Has The Least Execution Cost?*

Tables 9 and 10 show that for all selection strategies the proposed crossover had less execution cost.

**7. Threats to Validity**

In this paper, we identified three types of threats to the validity of our experimental study: internal validity, external validity, and construct validity. Internal validity refers to the cause-and-effect relationships, the extent to which we can state that the changes in dependent variables are caused by changes in independent variables [53]. Two internal threats to validity are as follows:

1. The setup and configuration of the parameters of the genetic algorithm. Identifying the optimal configuration that may lead to the best results can be hard. Moreover, the performance of the tool may vary from one machine to another as the idleness of the machine cannot be guaranteed. However, in this paper we performed an experimental evaluation to identify the best configuration for the algorithm. We also ran the tool on an isolated environment to insure that the machine is not running anything besides the tool. Moreover, to minimize this effect, we performed each run 10 times.
2. The number and quality of the test cases. We used three different tools to generate the test cases. Using other tools may lead to different results. However, the test cases used had a 100% branch coverage and were able to kill all the generated FOMs. Moreover, we used handcrafted test cases to ensure the quality of the test cases when the test cases generated by the tools failed to cover all branches or kill all FOMs.

External validity is concerned with generalizing the results outside the scope of the experiment [53]. Two external threats to validity are as follows:

1. The subject programs. We performed the empirical evaluation on 10 subject programs, and there is no evidence that the results can be extended or generalized to other Java programs or programs implemented in other programming languages. However, as mentioned earlier, the selected subject programs differ in their size, complexity, operations, and object-oriented programming concepts used.

2. The subject programs are small (less than 200 lines of code) and constituent from only one class. Using larger programs or more than class may produce different results. In the future, additional programs with larger sizes will be studied.

Construct validity refers to the meaningfulness of measurements [53]. One threat to construct validity comes from the tools used in the experiment, especially muJava and GaSubtle, which generate the mutants. To minimize this threat, we inspected randomly selected outputs and manually verified their correctness. Moreover, the manual identification of equivalent mutants is considered as another threat to construct validity.

## 8. Conclusions

In this paper, a new approach for generating subtle higher-order mutants is developed. The approach uses a genetic algorithm along with five selection strategies: *Roulette Wheel, Stochastic Universal Sampling, Tournament, Truncation and Random Selection*. We also suggested a new mechanism for crossover. The proposed crossover is performed by selecting the most fit FOMs from both parents and placing them in the first child, and the least fit FOMs in the second child. If the two parents differ in order, then the most fit children are placed in the lower-order mutant. We also developed a tool called GaSubtle based on Java and JavaFX that implements the proposed approach.

In order to evaluate the effectiveness of the proposed approach, an experimental study is performed by using 10 subject programs. A test suite is generated by using various tools for each program. The test suite satisfies branch coverage and kills all of the FOMs of the subject program (excluding equivalent mutants). The results of the experiment can be summarized as follows:

**Equivalent mutants**: Both single-point and proposed crossover generated a reasonable number of equivalent mutants (less than 7%). However, the proposed crossover generally generates less equivalent mutants compared with single-point crossover. For selection strategies, truncation and tournament selection generated the minimum percentage of equivalents mutants, and random selection generated the maximum percentage.

**Subtle mutants**: The proposed crossover generated significantly more subtle mutants compared with single-point crossover. For selection strategies, truncation selection generated the maximum number of subtle mutants.

**Mutation order**: For both single-point crossover and the proposed crossover, the majority of the generated mutants were in the second order. It may be due to creating an initial population with an 80% of SOMs, or because adding more mutants to an HOM makes it easier to be killed. This result however, confirms with the findings of earlier studies, which show that most of the subtle mutants are in the lower order (e.g., [37]).

For future work, we plan to perform more experimental studies in order to understand the relation between the program under test and the selection strategy. We also plan to enhance the GaSubtle tool by allowing the tool to work with programs developed in programming languages other than Java.

**Author Contributions:** Conceptualization, F.W.; methodology, F.W. and A.A.-S.; software, A.A.-S. and F.W.; validation, F.W., A.A.-S. and Y.J.; formal analysis, F.W. and A.A.-S.; investigation, F.W., A.A.-S. and Y.J.; resources, F.W., A.A.-S. and Y.J.; data curation, F.W., A.A.-S. and Y.J.; writing—original draft preparation, F.W., A.A.-S. and Y.J.; writing—review and editing, F.W., A.A.-S. and Y.J.; visualization, F.W., A.A.-S. and Y.J.; supervision, F.W.; project administration, F.W.; funding acquisition, Not applicable. All authors have read and agreed to the published version of the manuscript.

**Funding:** This research received no external funding.

**Institutional Review Board Statement:** Not applicable.

**Informed Consent Statement:** Not applicable.

**Data Availability Statement:** Results can be regenerated using the available tool GaSubtle.

**Conflicts of Interest:** The authors declare no conflict of interest.

## Abbreviations

The following abbreviations are used in this manuscript:

| | |
|---|---|
| FOM | First Order Mutant |
| SOM | Second Order Mutant |
| HOM | Higher Order Mutant |
| GA | Genetic Algorithm |
| LOC | Lines Of Code excluding spaces |
| MVC | Model View Controller |
| GaSubtle | Genetic algorithm for generating subtle higher order mutants |

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
