# Peer review of "GaSubtle: A New Genetic Algorithm for Generating Subtle Higher-Order Mutants"

_information, doi:10.3390/info13070327_

Round 1

Reviewer 1 Report

The main contribution of this manuscript is introducing a tool for generating HOMs. The manuscript claims those HOMs are subtle mutants. this manuscript needs a comprehensive revision to be accepted in a good Journal. the following comments can help in improving the quality of this manuscript.

there are many English writing typos.

FOM needs to be defined before use.

the objectives of the manuscript must be clearly written at the end of the introduction. 

the related work must focus on the problem of the manuscript.

the representation of the chromosome is not clear.

what is the ref of equation 4?

LOC is not the only reason for the number of generated mutants.

how the mutation process can be applied to mutants of order greater than second-order?

The crossover process is not clear.

how the fitness of FOM can be found to it in the crossover.

the structure of the given example is not readable.

how the layer pattern can help in the reusability and portability of the proposed tool.

what are the relations between layers in figure 2?

the set of subject programs is very small; another experiment with bigger programs is needed.

the proposed tool will face the explosion problem how can solve this problem

the conclusion is very long 

there is a huge number of references 86; 30-40 are enough.

what is the significance of the proposed method?

Author Response

Dear reviewer:

Thank you for agreeing to review or work, below we give our response for your comments:

The main contribution of this manuscript is introducing a tool for generating HOMs. The manuscript claims those HOMs are subtle mutants. this manuscript needs a comprehensive revision to be accepted in a good Journal. the following comments can help in improving the quality of this manuscript.

  1. There are many English writing typos.

Response: We revised the paper, and did our best to find all typos.

  1. FOM needs to be defined before use.

Response: fixed in the introduction, line 61

  1. the objectives of the manuscript must be clearly written at the end of the introduction. 

Response: we stated the objectives clearly as indicated. Please refer to lines 93-105

  1. the related work must focus on the problem of the manuscript.

Response: We revised the related work section, we kept work that on higher order mutation testing

  1. the representation of the chromosome is not clear.

Response: Clarified

  1. what is the ref of equation 4?

Response: there should be no reference for eq 4. We developed this formula to set a general rule for the size of the generation. We changed the text to clarify that.

  1. LOC is not the only reason for the number of generated mutants.

Response: We agree, we just used this factor in determining the generation size. The formula we used helps in finding a suitable generation size to speed up the algorithm. We rephrased the text to show that (lines 323-326)

  1. how the mutation process can be applied to mutants of order greater than second-order?

Response: by adding or removing a FOM, the exception is just the second order, in which we only add, to ensure that the resulting mutant will not be a FOM. The text is rephrased to clarify the process (lines 361-364)

  1. The crossover process is not clear.
  2. how the fitness of FOM can be found to it in the crossover.

Response: The tool computes it, we clarified this detail. It’s also step one of the algorithm

  1. the structure of the given example is not readable.

Response: for these three comments, we rewrote the crossover section to clarify the raised concerns.

  1. how the layer pattern can help in the reusability and portability of the proposed tool.

Response: layered architecture supports these two quality attributes by providing a physical layer (can have many names) to deal with different platforms (and therefore supporting portability), and by having layers which communicate by method invocation, modularity and therefore, reusability is increased.

However, we clearly do not need to discuss these issues in the paper so, we just mentioned portability.   

  1. what are the relations between layers in figure 2?

Response: method invocation(requests) are send from a layer to its underneath layer. We clarified this detail in the tool section.

  1. the set of subject programs is very small; another experiment with bigger programs is needed.

Response: For the above two comments, we know that our empirical evaluation has limitations which we discussed in Section 7 (Threats to Validity). A major part in our future work is to conduct a large empirical evaluation on relatively big projects. This will be a separate study, in the current study, we developed a new approach, we developed a tool for the approach and performed evaluation on a limited set of programs.

  1. the proposed tool will face the explosion problem how can solve this problem

Response: This might happen, we might consider using multi-threads in version 2 of the tool. Currently we are trying larger programs, but so far the performance is reasonable (in minutes).

  1. the conclusion is very long 

Response: we shortened the conclusions

  1. there is a huge number of references 86; 30-40 are enough.

Response: We shortened the list, we shortened related work by keeping only HOM studies. We also used footnotes to cite webpages and the subject programs.

  1. what is the significance of the proposed method?

Response: Solving the problems of mutation testing can ease adopting the approach by the industry.

Reviewer 2 Report

This paper proposes a new genetic algorithm to generate subtle higher order mutants. The algorithm is developed a tool named GaSubtle which based on five selection strategies and a new crossover method. The experimental results show the effectiveness and low cost of the algorithm.

 Pros.

1. The proposed crossover method is novel, which is performed to generate next population by selecting the most fit FOMs from both parents and placing them in the first child, and the least fit FOMs in the second child.

 2. Compared with the single point crossover, the authors conduct some RQs and experiments to demonstrate the effectiveness and low cost of the proposed crossover method.

 3. The authors have developed a user-friendly tool.

 Cons.

1. The new genetic algorithm proposed by the paper aims to generate subtle HOMs, but the design of the experiments is confused and make the demonstration of the algorithm's effectiveness is unclear.

 2. In the paper, there are some inaccurate and inconsistent descriptions.

 3. In the paper, the analysis of some experimental results is missed.

Comments.

1. About the RQs, the single point crossover as compared method, but the related description is missed. So, what is the single point crossover and the difference that between single point crossover and proposed crossover? In Section 2, the authors introduce some related works, why not use the methods in the related works as the compared method and what is the reason for choosing the single point crossover?

2. In Section 6. There are some inconsistence and wrong descriptions in the sub-Section about the RQs. Such as the RQ3 in Section 6.3 and Section 5.2 is inconsistent; the RQ5 in Section 6.5 and the RQ6 in Section 6.6 is different with the Section 5.2; the RQ in Section 6.6 and Section 6.7 is the same one. In addition, the RQ1 and RQ2 use the same experiment method and results, the RQ3 and RQ4 is the same way. So, it’s may be to merge the RQs which used the same method and results.

3. About the results and analysis in Section 6, I find that most results are a list of results, and the analysis of the results is missed. Such as, the reason for the results in Section 6.3.1; In Section 6.4, the analysis of the result that the single point crossover generated equivalent mutants slightly less than the proposed crossover on truncation selection; In Section 6.5, the reason for the difference between high order mutants and low order mutants in generating HOMs.

4. In the Section 3.5. The proposed crossover’s description “GaSubtle performs crossover by selecting the most fit FOMs from both parents and placing them in the first child, and the least fit FOMs in the second child” and the example’s description (line 452 and line 480) is inconsistent and confused. Is the high value not the most fit? If yes, why the low fitness is placed in child and the high fitness is placed in child 2? At the same Section, the second sentence descript processed method of the two parents differ in order. How about the processed method when two parents same in order?

5. In Section 5.2, there are seven RQs. But Section 1 descripts three questions. What is the reason for doing this? If the Section 1 is the summary for the Section 5.2, how it is summarized. In addition, the order and description logic of the 7 RQs is confused.

6. In Section 5.3.2. The description “In this experiment the following tools are used:” (line 663) and “In this experiment Jtest was used to generate test cases only” (line 675) is inconsistent.

7. In Section 5.4.2, paragraph 1, last sentence. The test cases generated by some tools, when the generated test cases did not satisfy two criteria and the test cases are added manually. So, In the Table 3, what are the sources of test cases for each project and how many test cases are added manually?

8. In Section 6.1.2, paragraph 1, sentence 3. There descript four subject programs while the Table 5 have five.

9. In Section 6.3, paragraph 1, sentence 2. What is the method of subtle HOMs examination?

10. In Section 6.5. Table 8 and Table 9 is the percentage value and the number should have %. In addition, Table 8 and Table 9 is about HOMs, but the second paragraph description is subtle mutants in this Section. Is it wrong or there are some reasons?

11. The keywords “search-based software testing” and “evolutionary algorithm” not in the Abstract.

12. In the line 82. This should be subtle HOMs in the sentence “we introduce a new genetic algorithm for generating HOMs using genetic algorithm”.

13. In Section 2. There are many related works, such as HOM testing, HOM, and subtle HOM. But it may be better for the reader to understand the related work if the author could list them in categories.

14. In Section 2, last paragraph. There is a literature citation problem.

15. In Section 7, paragraph 2, sentence 2. There is a repeat word “threat”.

Author Response

Dear reviewer:

Thank you for agreeing to review or work, below we give our response for your comments:

 Pros.

  1. The proposed crossover method is novel, which is performed to generate next population by selecting the most fit FOMs from both parents and placing them in the first child, and the least fit FOMs in the second child.
  2. Compared with the single point crossover, the authors conduct some RQs and experiments to demonstrate the effectiveness and low cost of the proposed crossover method.
  3. The authors have developed a user-friendly tool.

 Cons.

  1. The new genetic algorithm proposed by the paper aims to generate subtle HOMs, but the design of the experiments is confused and make the demonstration of the algorithm's effectiveness is unclear.
  2. In the paper, there are some inaccurate and inconsistent descriptions.
  3. In the paper, the analysis of some experimental results is missed.

Comments.

  1. About the RQs, the single point crossover as compared method, but the related description is missed. So, what is the single point crossover and the difference that between single point crossover and proposed crossover? In Section 2, the authors introduce some related works, why not use the methods in the related works as the compared method and what is the reason for choosing the single point crossover?

Response: The single point crossover is used by many previous studies, that why we choose to compare with it. We clarified this.

  1. In Section 6. There are some inconsistence and wrong descriptions in the sub-Section about the RQs. Such as the RQ3 in Section 6.3 and Section 5.2 is inconsistent; the RQ5 in Section 6.5 and the RQ6 in Section 6.6 is different with the Section 5.2; the RQ in Section 6.6 and Section 6.7 is the same one. In addition, the RQ1 and RQ2 use the same experiment method and results, the RQ3 and RQ4 is the same way. So, it’s may be to merge the RQs which used the same method and results.

Response: We fixed the mismatch, one of the questions (question 5) was dropped from the empirical study but we forgot to delete it. RQ1 and RQ2 depend on the same results but the answer for each give two different results. We updated the text to clarify the results.

  1. About the results and analysis in Section 6, I find that most results are a list of results, and the analysis of the results is missed. Such as, the reason for the results in Section 6.3.1; In Section 6.4, the analysis of the result that the single point crossover generated equivalent mutants slightly less than the proposed crossover on truncation selection; In Section 6.5, the reason for the difference between high order mutants and low order mutants in generating HOMs.

Response: We revised the results section, we added our interpretation of the results. We hope it’s more clear now.

  1. In the Section 3.5. The proposed crossover’s description “GaSubtle performs crossover by selecting the most fit FOMs from both parents and placing them in the first child, and the least fit FOMs in the second child” and the example’s description (line 452 and line 480) is inconsistent and confused. Is the high value not the most fit? If yes, why the low fitness is placed in child and the high fitness is placed in child 2? At the same Section, the second sentence descript processed method of the two parents differ in order. How about the processed method when two parents same in order?

Response: if the two parents have the same order, one will have the fittest FOMs and the other will have the least fit. We added text to clarify this, in step 2 of the crossover procedure. Thank you

  1. In Section 5.2, there are seven RQs. But Section 1 descripts three questions. What is the reason for doing this? If the Section 1 is the summary for the Section 5.2, how it is summarized. In addition, the order and description logic of the 7 RQs is confused.

Response: We changed the questions so that they match (Section 5.2 and Section 6). In the introduction, we have 3 questions as the questions are stated abstractly, which then been extracted to 6 questions.

  1. In Section 5.3.2. The description “In this experiment the following tools are used:” (line 663) and “In this experiment Jtest was used to generate test cases only” (line 675) is inconsistent.

Response: In line 675, what we meant is that we did not use all functionalities of Jtest, we just used it for generating test cases. The line is confusing as it looks, we removed the confusing statement.

  1. In Section 5.4.2, paragraph 1, last sentence. The test cases generated by some tools, when the generated test cases did not satisfy two criteria and the test cases are added manually. So, In the Table 3, what are the sources of test cases for each project and how many test cases are added manually?

Response: None of the test cases are manually created. In this experiment, we did not face this problem. But the procedure says “in case” the tools failed to create the test cases, we add manually.

  1. In Section 6.1.2, paragraph 1, sentence 3. There descript four subject programs while the Table 5 have five.

Response: We double checked Table 5, there are only 4 programs in which truncation selection generated the highest number of subtle mutants

  1. In Section 6.3, paragraph 1, sentence 2. What is the method of subtle HOMs examination?

Response: The answer is given in the following line “The examination was done by inspection to a random 100 mutants of the generated subtle HOMs for each selection strategy. However, some of the selection strategies generated less than a 100 subtle HOMs, in such cases, all of the generated subtle HOMs were examined.”

  1. In Section 6.5. Table 8 and Table 9 is the percentage value and the number should have %. In addition, Table 8 and Table 9 is about HOMs, but the second paragraph description is subtle mutants in this Section. Is it wrong or there are some reasons?

Response: We should made it clear, the question is about subtle HOM.

  1. The keywords “search-based software testing” and “evolutionary algorithm” not in the Abstract.

Response: We do not know that all key words have to be mentioned in the abstract. The field of the study is search-based software testing and the solution is a genetic algorithm, which is an evolutionary algorithm. But if the reviewer thinks the keywords should be changed, we can do that

  1. In the line 82. This should be subtle HOMs in the sentence “we introduce a new genetic algorithm for generating HOMs using genetic algorithm”.

Response: Corrected

  1. In Section 2. There are many related works, such as HOM testing, HOM, and subtle HOM. But it may be better for the reader to understand the related work if the author could list them in categories.
  2. In Section 2, last paragraph. There is a literature citation problem.

Response: Corrected

  1. In Section 7, paragraph 2, sentence 2. There is a repeat word “threat”.

Response: Corrected

Reviewer 3 Report

The paper in its present form should be rejected. However, after fixing all the serious defects and methodological flaws, the paper could be considered for a publication.

The main issue I have with this research is that the authors introduce some new solution for generating subtle Higher Order Mutations, but they don't even compare it with the existing ones described in Section 2. So we don't know if this solution is even better than any other solution already proposed. This is the main and very serious problem with this paper.

The authors did not provide the additional material that would allow the reviewer to verify the expeirments. The code should be publicly available (at least for the reviewers). This is nowadays the standard in the empirical software engineering.

Also, the paper is written in a very bad English. The grammar, stylistic and interpunction errors are almost in each paragraph. Below I present only some of them, from first two or three pages - I didn't comment on the rest of them, since the revision would be very long.

Other major remarks:

The paper uses the term 'genetic algorithm', but what the authors really use is *evolutionary* algorithm, since (simple) genetic algorithm operates on binary chromosomes, which is not the case here.

79 - it is not true that the optimal solution cannot be found for the NP-complete problem! If P != NP we can say that no POLYNOMIAL time algorithm cannot be found. There are exponential algorithms that find optimal solutions to NPC problems

125-127 - It is not true that HOMs subsume FOMs. For example, consider the very simple example:
input x
y := x + 1
y := x - 1
return y
if FOM changes + to -, we may kill the mutant. owever, HOM in which + is changed to - and - to + gives the equivalent mutant.

263 - missing reference

294 - notational collision - don't use "t" as both a test case and its index ("t_{c_t}")

296 - add "for some i_1, ..., i_n". Add also that they should be different (I assume you want to have here exactly n different FOMs)

300-301 - a notation problem; in T_{i_j} you should use h_i, since you won't be able to differ T_{i_j} sets for different h_i (for each h_i the corresponding indices will be the sae: i_1, i_2 and so on) - however, see the next remark, since this remark may be irrelevant

300-301 - what do you mean by "to kill f_{i_j} in h_i"? How can a FOM be killed "in" a HOM? Is it possible that for some HOM consisting of two FOMs one of these FOMs will be killed by some test and another one won't? Isn't the "killing" concept for HOM defined in a way that a test either kills HOM or not (?). If so, also the notion of TU_i makes no sense, since it will be always equal to T_{h_i}. I suppose in line 301 there should be just "that kill f_{i_j}", without "in h_i". But then, the notion of TU_i also makes no sense, since it is always equal to T

Please give a concrete example of HOM so that the reader will be able to understand how HOMs are created out of FOMs. Please also interpret the FDD and DOK measures: what does it mean if they equal to 0? Whar does it mean when they equal to 1? Provide a simple, but non-trivial example with these metrics (a set of tests, FOMs and one HOM for which these metrics have values greater than 0 but less than 1.

eq 1 and 2 - hi -> h_i

eq 2 - no parentheses needed in |(T_hi)|

312 - please be more specific what do you mean by a solution. From te context it is not clear that a solution represents only one HOM.

322-325 - what kind of mutation operators did you use? This will have the obvious impact on the number of possible, potential mutants. Did you use for example object-oriented mutation operators or just simple low-level code mutations? Please list all the operators used

329 - did you mean non-comment lines of code? The present formulation is weird, because it excludes lines that contain spaces (?!). Please use a well-defined metric here, for example above mentioned NCLOC or executable LOC or sth similar, but precise enough.

356-363 - please discuss this in detail. You only said that for 2nd order HOM mutation can only add FOM; what about 3rd and 4th order? Can we make 2nd order HOM from 3rd order HOM? And from 4th order as well? Can we add FOM to 4rd order HOM to create 5th order HOM? Please describe the rules precisely.

365 - you write about "most fit FOMs" but you don't write earlier, that FOMs "isolated" from considered HOMs can be also subject to the fitness function. The fitness is calculated for population elements, which are (usually) HOMs.

Fig. 1 - what do the numbers like 0.5, 0.9 and so on mean? Are these some probabilities of exchange? Or the fitness calculated for the "isolated" FOMs? Be more specific

365-369 + figure 1 - it is still unclear for me how does it work. Which FOMs are chosen to be exchanged? Why the fitness for the whole HOM does not matter here? What do you mean by "lower order" for children? This is completely unintelligible.

please use some nice latex environment like "algorithm2e" so that the code looks nice; now it is very badly edited

The example of HOMs and their crossover should be presented in (preferably) one Figure, so that it's easier to read and understand; the mutated places should be denoted with some colors

512 - the criterion "reaching a given number of subtle HOMs" is vague, because you did not define precisely which mutants are subtle (you just write at the beginning that these are the ones that are "harder" to be killed, which is no definition at all. This is also important for line 15 of the algorithm. Also, line 512 is gramatically incorrect

algorith 1 - does every line have to be in italic?... it's difficult to read

general remark about the tool description: technical details can be omitted or put in the appendix, they are not releveant here, since the most important thing is to show the empirical evaluation of the algorithm regarding generating subtle HOMs, it's not a user documentation for the software

Programs in your analysis are *very* small and simple. The study should be more pragmatic, you should use some more real-life scenarios and algorithms. Otherwise, the whole framework and the experiment is pointless, since in practice people work with much complicated code

585-591 - wouldn't the use of byte code be useful here? You could operate directly on bytecode, not the source code, hence there wouldn't be even a need to compile the code (or at least, the number of compilations would be much smaller)

604 - you say the selected programs "contain various constructs and solve different problems", but in fact they are all very similar - they are some simple math procedures or some simple, classical text algorithms. So they are all similar! They don't even differ in size. 

714 - taking 10 runs of the experiment is far too small number, considering that evolutionary algorithms are based on randomness and considering the number of possible solutions. The variance in the measured metrics may be very high, so there is a high risk of having large measurement errors.

General remark for the experiments: with random-based algorithms you ave to use statistical inference to interprete the results. If you compare two approaches (for example, two different selection strategies) you have to use tools like Student's t-test to verify if the obtained differences are statistically significant. It may happen that the obtained differences are due to pure randomness, and not due to the measured factor!

tables 10 and 11 - time in minutes should be at least given with 0.1 precision

Minor remarks:

3 - better: "equivalent mutants, which are faulty (...)"
6, 7 - bad style of the sentence
10 - elect -> select
10 - goes -> go
11 - that automates -> what does it automate??
15, 16 - separate keywords with semicolons, not commas
21 - "decided" - this word does not go here, better: evaluated, related to etc.
31, 32 - bad style - you write about two problems, but describe one only
41 - suggest -> suggests
44 - add space before [9]
46 - add space before [10]
51 - minimizes -> minimize
60 - use the full name for FOM when you use it first time
66 - have -> has
69 - double citation of [6]
72 - hide -> mask
72 - "not each..." etc. - bad style and grammatically incorrect
79 - an optimal -> the optimal
122-123 - the sentence has bad style (and no verb)
128 - change comma to period
134 - in -> is
144-146 - the sentence has no verb
146 - passes -> passed or pass

Author Response

Dear reviewer:

Thank you for agreeing to review or work, below we give our response for your comments:

The paper in its present form should be rejected. However, after fixing all the serious defects and methodological flaws, the paper could be considered for a publication.

The main issue I have with this research is that the authors introduce some new solution for generating subtle Higher Order Mutations, but they don't even compare it with the existing ones described in Section 2. So we don't know if this solution is even better than any other solution already proposed. This is the main and very serious problem with this paper.

Response: We compared our approach with an existing approach of crossover, which is used in previous works.

The authors did not provide the additional material that would allow the reviewer to verify the experiments. The code should be publicly available (at least for the reviewers). This is nowadays the standard in the empirical software engineering.

Response: The tool is available on Github. It’s our mistake as we didn’t mention that. We added a link to the tool in the text

https://github.com/AbdullahAsendar/GaSubtle

Also, the paper is written in a very bad English. The grammar, stylistic and interpunction errors are almost in each paragraph. Below I present only some of them, from first two or three pages - I didn't comment on the rest of them, since the revision would be very long.

Response: We performed more than one round of corrections. We hope the paper looks better now.

Other major remarks:

The paper uses the term 'genetic algorithm', but what the authors really use is *evolutionary* algorithm, since (simple) genetic algorithm operates on binary chromosomes, which is not the case here.

Response: Thank you for bringing up this issue. Genetic algorithm is a sub-class of evolutionary algorithm, but since the selection is probabilistic and we use crossover as the main method for reproduction, then we believe it’s a genetic algorithm as it follows the steps of the algorithm developed by Holland. If we say just evolutionary algorithm, then we need to specify what type of evolutionary algorithms it is (GA, evolutionary programming, or evolution strategy), which will lead us to saying it’s a genetic algorithm.

79 - it is not true that the optimal solution cannot be found for the NP-complete problem! If P != NP we can say that no POLYNOMIAL time algorithm cannot be found. There are exponential algorithms that find optimal solutions to NPC problems

Response: We apologize for this incorrect info, the text is modified.

125-127 - It is not true that HOMs subsume FOMs. For example, consider the very simple example:
input x
y := x + 1
y := x - 1
return y
if FOM changes + to -, we may kill the mutant. owever, HOM in which + is changed to - and - to + gives the equivalent mutant.

Response: We didn’t say that HOMs subsume FOMs. What Jia is referring to is a subset of HOM called “subsuming HOM” that has this property. We rewrote this part to make it clear.

263 - missing reference

Response: Corrected

294 - notational collision - don't use "t" as both a test case and its index ("t_{c_t}")

Response: corrected. We used the same format used by the authors of the fitness function (which has this mistake).

296 - add "for some i_1, ..., i_n". Add also that they should be different (I assume you want to have here exactly n different FOMs)

Response: Corrected

300-301 - a notation problem; in T_{i_j} you should use h_i, since you won't be able to differ T_{i_j} sets for different h_i (for each h_i the corresponding indices will be the same: i_1, i_2 and so on) - however, see the next remark, since this remark may be irrelevant

300-301 - what do you mean by "to kill f_{i_j} in h_i"? How can a FOM be killed "in" a HOM? Is it possible that for some HOM consisting of two FOMs one of these FOMs will be killed by some test and another one won't? Isn't the "killing" concept for HOM defined in a way that a test either kills HOM or not (?). If so, also the notion of TU_i makes no sense, since it will be always equal to T_{h_i}. I suppose in line 301 there should be just "that kill f_{i_j}", without "in h_i". But then, the notion of TU_i also makes no sense, since it is always equal to T

Response: We added a text explaining all these issues.

Please give a concrete example of HOM so that the reader will be able to understand how HOMs are created out of FOMs. Please also interpret the FDD and DOK measures: what does it mean if they equal to 0? Whar does it mean when they equal to 1? Provide a simple, but non-trivial example with these metrics (a set of tests, FOMs and one HOM for which these metrics have values greater than 0 but less than 1.

eq 1 and 2 - hi -> h_i

eq 2 - no parentheses needed in |(T_hi)|

Response: We added a text explaining the terms FDD and DOK, we also explains how fitness is interpreted using the chosen function. The formulas are also corrected.

312 - please be more specific what do you mean by a solution. From te context it is not clear that a solution represents only one HOM.

Response: a solution is one HOM. We clarified this

322-325 - what kind of mutation operators did you use? This will have the obvious impact on the number of possible, potential mutants. Did you use for example object-oriented mutation operators or just simple low-level code mutations? Please list all the operators used.

Response: we used all mutation operators implemented in muJava. We mentioned that in the tool section. We also updated in the text in lines 322-325

329 - did you mean non-comment lines of code? The present formulation is weird, because it excludes lines that contain spaces (?!). Please use a well-defined metric here, for example above mentioned NCLOC or executable LOC or sth similar, but precise enough.

Response: We counted lines of source code, ignoring comments and empty lines. The text is updated.

356-363 - please discuss this in detail. You only said that for 2nd order HOM mutation can only add FOM; what about 3rd and 4th order? Can we make 2nd order HOM from 3rd order HOM? And from 4th order as well? Can we add FOM to 4rd order HOM to create 5th order HOM? Please describe the rules precisely.

Response: we updated the text. All mutants can be mutated by adding or removing a FOM. The exception is second order mutants, for which we only use adding, because we do not want to have FOMs in the population

365 - you write about "most fit FOMs" but you don't write earlier, that FOMs "isolated" from considered HOMs can be also subject to the fitness function. The fitness is calculated for population elements, which are (usually) HOMs.

Fig. 1 - what do the numbers like 0.5, 0.9 and so on mean? Are these some probabilities of exchange? Or the fitness calculated for the "isolated" FOMs? Be more specific

365-369 + figure 1 - it is still unclear for me how does it work. Which FOMs are chosen to be exchanged? Why the fitness for the whole HOM does not matter here? What do you mean by "lower order" for children? This is completely unintelligible.

Response: the text is updated to clarify these concerns.

please use some nice latex environment like "algorithm2e" so that the code looks nice; now it is very badly edited

Response: We changed the environment used for the code. Hopefully it is better now

The example of HOMs and their crossover should be presented in (preferably) one Figure, so that it's easier to read and understand; the mutated places should be denoted with some colors

Response: The examples are updated as requested (in figures), and with better explanation.

512 - the criterion "reaching a given number of subtle HOMs" is vague, because you did not define precisely which mutants are subtle (you just write at the beginning that these are the ones that are "harder" to be killed, which is no definition at all. This is also important for line 15 of the algorithm. Also, line 512 is gramatically incorrect

algorith 1 - does every line have to be in italic?... it's difficult to read

Response: We found the algorithmic package option to change the italic style (which is the default). It’s easier to read now. Thanks

general remark about the tool description: technical details can be omitted or put in the appendix, they are not releveant here, since the most important thing is to show the empirical evaluation of the algorithm regarding generating subtle HOMs, it's not a user documentation for the software

Response: We simplified (abstracted) the tool description. These details are irrelevant as mentioned. All implementation details are available on github

Programs in your analysis are *very* small and simple. The study should be more pragmatic, you should use some more real-life scenarios and algorithms. Otherwise, the whole framework and the experiment is pointless, since in practice people work with much complicated code.

585-591 - wouldn't the use of byte code be useful here? You could operate directly on bytecode, not the source code, hence there wouldn't be even a need to compile the code (or at least, the number of compilations would be much smaller)

604 - you say the selected programs "contain various constructs and solve different problems", but in fact they are all very similar - they are some simple math procedures or some simple, classical text algorithms. So they are all similar! They don't even differ in size. 

714 - taking 10 runs of the experiment is far too small number, considering that evolutionary algorithms are based on randomness and considering the number of possible solutions. The variance in the measured metrics may be very high, so there is a high risk of having large measurement errors.

General remark for the experiments: with random-based algorithms you have to use statistical inference to interprete the results. If you compare two approaches (for example, two different selection strategies) you have to use tools like Student's t-test to verify if the obtained differences are statistically significant. It may happen that the obtained differences are due to pure randomness, and not due to the measured factor!

Response: For the above two comments, we know that our empirical evaluation has limitations which we discussed in Section 7 (Threats to Validity). A major part in our future work is to conduct a large empirical evaluation on relatively big projects. This will be a separate study, in the current study, we developed a new approach, we developed a tool for the approach and performed evaluation on a limited set of programs.

tables 10 and 11 - time in minutes should be at least given with 0.1 precision

Response: corrected

Minor remarks:

3 - better: "equivalent mutants, which are faulty (...)"

6, 7 - bad style of the sentence

10 - elect -> select
10 - goes -> go
11 - that automates -> what does it automate??
15, 16 - separate keywords with semicolons, not commas
21 - "decided" - this word does not go here, better: evaluated, related to etc.
31, 32 - bad style - you write about two problems, but describe one only
41 - suggest -> suggests
44 - add space before [9]
46 - add space before [10]
51 - minimizes -> minimize
60 - use the full name for FOM when you use it first time
66 - have -> has
69 - double citation of [6]
72 - hide -> mask
72 - "not each..." etc. - bad style and grammatically incorrect
79 - an optimal -> the optimal
122-123 - the sentence has bad style (and no verb)
128 - change comma to period
134 - in -> is
144-146 - the sentence has no verb
146 - passes -> passed or pass

Response to Minor remarks: We corrected these issues in the text. Thank you

Round 2

Reviewer 1 Report

All references need to be formatted correctly see ref 54.

The conclusion is so long.

the abstract must include a summary of the obtained results.

Author Response

Dear Reviewer,

Thank you for your effort in reviewing our submission. The mentioned issues have been fixed. 

Reviewer 2 Report

The paper adds many detailed descriptions for the proposed method, such as the detailed description of the FDD and DOK that constitutes the fitness function in Section 3.1, the implementation details of the variation in Section 3.3, the specific steps of the crossover in Section 3.4 and so on. It makes the method clearer and more specific.

There are still some problems:

1. In Section 3.5, paragraph 3, sentence 2. There is a figure citation problem.

2. In Section 3.5. Figure 6 is not Child 2. It is the same as Figure 5.

3. In Section 6.7. RQ7 is not included in the research questions in Section 5.2.

Author Response

(The authors gave the same response as above.)
